# TNF-α-Mediated Endothelial Cell Apoptosis Is Rescued by Hydrogen Sulfide

**DOI:** 10.3390/antiox12030734

**Published:** 2023-03-16

**Authors:** Lorena Diaz Sanchez, Lissette Sanchez-Aranguren, Keqing Wang, Corinne M. Spickett, Helen R. Griffiths, Irundika H. K. Dias

**Affiliations:** 1Aston Medical School, College of Health and Life Sciences, Aston University, Aston Triangle, Birmingham B4 7ET, UK; 2School of Biosciences, College of Health and Life Sciences, Aston University, Aston Triangle, Birmingham B4 7ET, UK; 3Swansea Medical School, Swansea University, Singleton Park, Swansea SA2 8PP, UK

**Keywords:** vascular dysfunction, hydrogen sulfide, oxidative stress, inflammation, mitochondrial function

## Abstract

**Highlights:**

**Abstract:**

Endothelial dysfunction is implicated in the development and aggravation of cardiovascular complications. Among the endothelium-released vasoactive factors, hydrogen sulfide (H_2_S) has been investigated for its beneficial effects on the vasculature through anti-inflammatory and redox-modulating regulatory mechanisms. Reduced H_2_S bioavailability is reported in chronic diseases such as cardiovascular disease, diabetes, atherosclerosis and preeclampsia, suggesting the value of investigating mechanisms, by which H_2_S acts as a vasoprotective gasotransmitter. We explored whether the protective effects of H_2_S were linked to the mitochondrial health of endothelial cells and the mechanisms by which H_2_S rescues apoptosis. Here, we demonstrate that endothelial dysfunction induced by TNF-α increased endothelial oxidative stress and induced apoptosis via mitochondrial cytochrome c release and caspase activation over 24 h. TNF-α also affected mitochondrial morphology and altered the mitochondrial network. Post-treatment with the slow-releasing H_2_S donor, GYY4137, alleviated oxidising redox state, decreased pro-caspase 3 activity, and prevented endothelial apoptosis caused by TNF-α alone. In addition, exogenous GYY4137 enhanced S-sulfhydration of pro-caspase 3 and improved mitochondrial health in TNF-α exposed cells. These data provide new insights into molecular mechanisms for cytoprotective effects of H_2_S via the mitochondrial-driven pathway.

## 1. Introduction

A healthy vascular endothelium maintains vascular homeostasis by regulating vascular tone, angiogenesis, platelet aggregation, inflammation and oxidative status [1]. Inflammatory activation and endothelial dysfunction are the initial steps in the pathogenesis of atherosclerosis and pose an elevated risk for cardiovascular complications [2]. Understanding the mechanisms leading to endothelial dysfunction and the relationship with inflammatory pathways may inform novel therapeutic strategies to treat vascular diseases.

Endothelial dysfunction is associated with a pro-inflammatory phenotype, mitochondrial dysfunction, and imbalance in the cellular redox steady-state [3,4]. Pro-inflammatory cytokines such as tumour necrosis factor-α (TNF-α) initiate and regulate endothelial activation and, ultimately, cell dysfunction that can lead to cardiac- and metabolic vascular complications [5]. Neutralising TNF-α therapies in inflammatory arthritis successfully suppressed endothelial dysfunction, apoptosis and mitochondrial damage in patients with cardiovascular comorbidities [6,7].

In recent years, hydrogen sulfide (H_2_S) has gained attention for its potential beneficial roles in several vascular conditions, preserving vascular wall integrity and vascular tone [8,9]. Furthermore, H_2_S is known to restore the redox balance in vascular beds, increasing the activity of reactive oxygen species (ROS)-scavenging enzymes [10]. Apart from its well-known role as an antioxidant, H_2_S has been shown to affect biological functions via protein S-sulfhydration, a post-translational modification of reactive cysteine residues in proteins that result in the conversion of -SH to an -SSH group, leading to the modification of protein activity [11,12,13,14].

In mammalian systems, H_2_S is synthesised by the transsulfuration pathway and mitochondrial cysteine catabolism pathway. In this regard, mitochondria play an important role in H_2_S metabolism via the mitochondrial sulfide oxidation pathway [15]. As a gaseous molecule, H_2_S can easily travel across membranes. H_2_S entering mitochondria is oxidised by the inter-mitochondrial membrane protein, sulfide quinone oxidoreductase (SQR), to generate persulfides that are later oxidised to short-lived sulfite (SO_3_^2−^), thiosulfate (S_2_O_3_^2−^) and sulfate (SO_4_^2−^) [15]. H_2_S is considered a toxic molecule at high concentrations as these oxidation products cause cytotoxic effects by altering mitochondrial membrane potential and disrupting cellular energy production [15]. Therefore, the therapeutic potential of H_2_S should be explored at a sub-toxic dose.

To date, most in vivo and in vitro studies have described the beneficial effects of H_2_S as a pre-treatment, where H_2_S is given before the systemic/cellular insult [16,17,18] or as a co-treatment [19,20], thereby preventing rather than revoking cell damage. Although the beneficial role of H_2_S as a post-treatment in the presence of an inflammatory setting has been shown before [19], the underlying mechanisms and mitochondrial-dependent effect on the vasculature are not fully known. Therefore, the present study used a slow H_2_S-releasing donor, GYY4137, to investigate the mechanism by which H_2_S protects endothelial cells from mitochondrial dysfunction and apoptosis during inflammation.

## 2. Materials and Methods

### 2.1. Cell Culture

Human umbilical vein endothelial cells (HUVECs) were obtained from PromoCell (#C12203, Leicestershire, UK). Cells were cultured (passages 1 to 4) in Endothelial Growth Cell Medium (EGM2 #C22111) supplemented with antibiotics (100 units/mL penicillin and 100 µg/mL streptomycin #P43333, Sigma-Aldrich, Dorsert, UK) at 37 °C with 5% CO_2_. Unless otherwise stated, HUVECs (1.5 × 10^4^ cells/well) were seeded into a 96-well plate in EGM2 supplemented medium and cultured overnight for experiments. Once 80–90% confluence was reached, cells were exposed to 1 ng/mL, 3 h TNF-α (#210-TA-005, R&D Systems, Oxford, UK-) dissolved in PBS with 0.1% BSA followed by treatment with 100 µM, 21 h GYY4137 (#SML2470, Sigma-Aldrich, Dorset, UK-) (Appendix A). The experimental set-up involved four groups (Control (cells treated with EGM2 media only without any treatments), TNF-α, GYY4137, and TNF-α+GYY4137).

### 2.2. Cell Viability Assay

Metabolic capacity of viable cells was measured using a fluorometric test (#G8080, CellTiter-BluePromega, Southampton, UK) according to the manufacturer’s instructions. Briefly, after treatments, the medium was discarded, and a fresh EGM2 medium containing 120 µL CellTiter-Blue solution at 37 °C was added and incubated for 3 h. The fluorescence intensity (Ex/Em: 560/590 nm) was measured in a microplate reader (TECAN, Spak Lom Ltd., Männedorf, Switzerland).

### 2.3. Tube Formation Assay

HUVECs were maintained in endothelial EGM2 medium at 37 °C and 5% CO_2_ conditions until the day before the experiment. Growth factor reduced Matrigel (#354230, Corning, Loughborough, UK) was thawed at 4 °C overnight, and 50 µL/well was added into a cooled sterile transparent 96-well plate on ice with a cooled sterile tip and incubated at 37 °C for 30 min to solidify. Cells (1 × 10^4^ cells/well) were seeded onto wells containing Matrigel in duplicates. Plates were incubated at 37 °C and 5% CO_2_ for 1 h. After treatments, phase-contrast images were captured at 4× magnification in each well using a Nikon eclipse Ts2 microscope. The total number of branches and the number of junctions in each image were determined using the ImageJ software Angiogenic plugin.

### 2.4. Measurement of Intracellular ROS Formation

The total cellular peroxide concentration was determined by using the fluorescent probe CM-H_2_DCFDA™ (#C6827, Themo Fisher Scientific, Loughborough, UK). After treatments, the medium was discarded. Fresh EGM2 medium containing 10 µM CM-H_2_DCFDA was incubated at 37 °C for 30 min in the dark and analysed using a fluorescent microscope (Ex/Em: 485/530 nm, Nikon, Melville, NY, USA).

### 2.5. Measurement of MitoSox Oxidation

HUVECs were seeded at a density of 2.5 × 10^5^ in a 6-well culture plate and incubated overnight. Once treatments were completed, cells were gently detached with trypsin-EDTA and cells resuspended in DMEM (#D6429, Sigma-Aldrich, Dorset, UK) free-serum medium with 5 µM MitoSox™ (#M36008, Thermo Fisher Scientific, Loughborough, UK) for 20 min at 37 °C in the dark. Next, HUVECs were pelleted by centrifugation at 200× *g* for 5 min, and the pellet was washed with PBS. Finally, cells were resuspended in 500 µL PBS, and each sample was analysed using 10,000 cellular events using the BD Accuri C6 Plus flow cytometer.

MitoSOX™ oxidation was analysed by fluorescence microscopy image analysis. After treatments, the medium was discarded, and a fresh EGM2 medium containing 5 µM MitoSOX™ Red was added. Cells were incubated at 37 °C for 30 min in the dark and analysed using a fluorescent microscope (Ex/Em: 510/580 nm, Nikon, Melville, NY, USA).

### 2.6. Assessment of Mitochondrial Membrane Potential (Δψm)

HUVECs were incubated with 5 µM of JC-1 (#T3168, Thermo Fisher Scientific, Loughborough, UK), incubated at 37 °C for 30 min in the dark, and analysed using a fluorescent microscope (green spectrum 529 nm, red spectrum 590 nm, Nikon, Melville, NY, USA). The intensity of the red/green fluorescence ratio was used to indicate mitochondrial membrane potential.

### 2.7. Detection of H_2_S

The intracellular H_2_S content was analysed using fluorescent probe SF7-AM (#748110, Sigma Aldrich, Dorset, UK). After treatments, fresh medium containing 2.5 µM SF7-AM was added and incubated at 37 °C for 30 min. DAPI (#ab228549, Abcam, Cambridge, UK), as a counterstain for nuclear morphology, was incubated at 37 °C for 10 min. Subsequently, cells were washed twice with phosphate-buffered saline (#806552, Sigma Aldrich, Dorset, UK,) and resuspended in fresh medium followed by fluorescence microscopy analysis (Ex/Em: 488/530 nm, Nikon, Melville, NY, USA).

### 2.8. Interleukin-6 (IL-6) Levels

At the end of incubation, supernatants were collected and centrifuged at 200 × *g* to remove debris. The levels of interleukin-6 (IL-6) were determined by an ELISA (#DY206-05, R&D systems, Oxford, UK) according to the manufacturer’s instruction.

### 2.9. ICAM-1 (Intercellular Adhesion Molecule 1) Levels

To detect ICAM-1 expression, 1 × 10^5^ cells were incubated with anti-ICAM-1 (anti CD54-APC) antibody (1:100, Invitrogen, Loughborough, UK, #17054942) or anti-IgG2b kappa APC isotype control (1:100, Invitrogen, Loughborough, UK, #124714442) at 4 °C for 1 h. Cells were analysed using BD Accuri C6 Plus flow cytometer.

### 2.10. Western Blotting

Cytosolic fractions were prepared from HUVECs using a Nuclear Extract kit (# 400100, Active Motif, Cambridge, UK) following the manufacturer’s instructions. Cytosolic proteins (10 µg) were resolved using SDS-PAGE, then transferred to a nitrocellulose membrane (#GE10600004, Amersham, Sigma Aldrich, Dorset, UK) and incubated with primary antibodies against; Cytochrome c (Cyto c) (1:500, Abcam, #ab90529), cleaved-caspase 3 (1:500, Cell signalling, Leiden, The Netherlands, Cat# 9661), caspase 3 (1:500, Cell signalling, Leiden, The Netherlands,#14220) and actin (1:2000, Abcam, Cambridge, UK #ab82241), followed by fluorescence-conjugated secondary antibodies (IRDye^®^ 800CW (anti-rabbit, #926-32211) and 680CW (anti-mouse, #926-32212) from LI-COR Biosciences (Cambridge, UK). Images were analysed with Image J software.

### 2.11. Flow Cytometry Analysis of Apoptosis

To analyse apoptosis, 1 × 10^5^ cells were stained with an Annexin V and Propidium Iodide (PI) according to Apoptosis Staining Kit instructions (#040914, BioLegend, Amsterdam, The Netherlands). In scatter plots, early apoptotic cells were identified as Annexin V positive and PI negative. At the same time, late apoptosis/necrosis was labelled for Annexin V and PI positive on a BD Accuri C6 Plus flow cytometer.

### 2.12. Caspase 3/7 Activity

HUVECs were seeded in a white-walled 96-well plate at 1.5 × 10^4^ cells/well density in EGM2 medium and cultured overnight. After treatments, the caspase activity assay was conducted using the luminescence-based Caspase-Glo 3/7 detection kit (#G8981, Promega, Southampton, UK) according to the manufacturer’s instructions. Briefly, Caspase-Glo 3/7 reagent (100 µL) was added to each sample and mixed using a plate shaker for 30 s. The plate was incubated at 37 °C in the dark for 1 h, followed by luminescence measurement in a microplate reader (TECAN, Spak Lom Ltd., Männedorf, Switzerland). The relative luminescence values were normalised to cell numbers. Data are represented as relative luminescence values.

### 2.13. Biotin Switch Assay

The detection of S-sulfhydration was carried out according to Cheung and Lau [21] with modifications. HUVECs (1 × 10^6^ cells) were seeded in T75 flask and grown overnight before treatment with TNF-α (1 ng/mL, 3 h) followed by GYY4137 (100 µM, 21 h) post-treatment at 37 °C with 5% CO_2_. ECs were homogenised in HEN buffer (250 mM Hepes-NaOH pH 7.7, 1 mM EDTA and 0.1 mM Neocuproine) supplemented with 100 µM deferoxamine and protease inhibitors and centrifuged at 13,000× *g* for 30 min at 4 °C. Protein samples (350 µg) were incubated with four volumes of blocking buffer (HEN buffer supplemented with 2.5% SDS and 20 mM S-methyl methanethiosulfonate (MMTS, #208795)) at 50 °C for 20 min with continuous vortexing in the dark to block free thiols (-SH). The MMTS was removed by pre-cold acetone precipitation at −20 °C for 1 h. After removal of acetone by centrifugation at 13,000× *g* at 4 °C for 10 min, the proteins were resuspended in HENS buffer (HEN buffer containing 1% SDS) and 4 mM biotin-N-[6-(biotinamido) hexyl]-3′-(2′-pyridyldithio) propinamide (HPDP) (#ab145614, Abcam) at 25 °C for 3 h in the dark. Biotinylated proteins were then precipitated by streptavidin-agarose beads (#29200, Thermo Fisher Scientific) (overnight at 4 °C with continuous mixing). The beads were washed five times with PBS (1×) and centrifuged at 5000× *g* for 15 s. The biotinylated proteins were incubated with elution buffer (20 mM Hepes-NaOH pH 7.7, 100 mM NaCl, 1 mM EDTA) with 1% β-mercaptoethanol at 37 °C for 30 min with shaking. The biotinylated protein samples in SDS-PAGE sample buffer (without β-ME) were heated at 95 °C for 5 min.

Protein samples were separated on 10% SDS-PAGE gels, transferred to a nitrocellulose membrane and incubated with anti-caspase 3 antibody (1:500, Cell signalling, Loughborough, UK) overnight at 4 °C. The next day, the membrane was soaked in fluorescence-conjugated secondary antibodies (IRDye^®^ 800CW (anti-rabbit, #926-32211, LI-COR Biosciences, Cambridge, UK). Revert™ 700 Total protein stain kit (#926-11010, LI-COR Biosciences, Cambridge, UK) was used for normalisation following manufacturer’s instructions.

### 2.14. Mitochondrial Network Assay

Mitochondrial morphology was determined by immunostaining followed by confocal microscopy, as described by Rao et al. [12]. Briefly, cells at 1.5 × 10^4^ density in EGM2 medium plate on 12 mm coverslips were treated with MitoTracker Red (100 nM) (#M7512, Thermo Fisher Scientific, Loughborough, UK), washed with PBS, fixed with 4% paraformaldehyde and stained with DAPI mounting medium (Invitrogen, Loughborough, UK). Images were collected by confocal lighting microscopy TCS SP8 system (Leica Microsystems Ltd., Milton Keynes, UK). Cells were analysed using an Ex/Em: 579/599 for Mitotracker and Ex/Em: 350/465 for DAPI. Images were acquired using a 63 × oil APO objective lens and analysed using Image J open-source macro tool, MINA (mitochondria network analysis) [12]. Settings were kept consistent across images and “mean” filter before filter was selected; a total number of 30 cells per group per individual experiment (n = 4) was consolidated and analysed for the network’s branching and branch length.

### 2.15. Real-Time PCR

Total RNA was isolated using the RNeasy Mini kit (Qiagen, Manchester, UK) and subjected to reverse transcription using Evo Scrip™ cDNA MasterMix Kit (#07912374001, Roche Diagnostic, Ltd., Welwyn Garden City, UK) following the manufacturer’s instructions. SYBR^®^ Green (#04707516001, Roche Diagnostic, Ltd., Welwyn Garden City, UK) RT-qPCR was performed using RT-qPCR system (LightCycler 480 II system, Roche Diagnostic, Ltd., Welwyn Garden City, UK). The relative mRNA levels were normalised to mRNA levels of EEF2, and calculation of each mRNA levels were made on comparative cycle threshold method (2^−ΔΔCt^). Primer-sequences used in this study are detailed in Appendix A.

### 2.16. Statistical Analysis

Data analysis was performed using Graph Prism (v. 8.0) software and expressed as means ± SD. Normality of the data was first assessed using Shapiro–Wilk normality test. Subsequently, one-way ANOVA analysis of variance followed by Tukey’s multiple comparison test or unpaired Student-t test (between two groups) was used. *p* < 0.05 was considered statistically significant. Experiments were performed independently at least triplicate.

## 3. Results

### 3.1. Exogenous H_2_S Ameliorates Intrinsic Apoptotic Pathways in Endothelial Cells

To test the ability of GYY4137 (100 µM) to increase intracellular H_2_S levels in our model, cells were stained with H_2_S-interacting dye, SF-7AM. As shown in Figure 1A,B, fluorescence staining confirmed that GYY4137 (100 µM) enhanced the intracellular H_2_S level in HUVECs (*p* < 0.005) without toxicity as demonstrated by mitochondrial superoxide production (Appendix A).

TNF-α i induces cell death by activating extrinsic and intrinsic apoptosis pathways [22,23]. Flow cytometry and ELISA analysis revealed that TNF-α treatment increases inflammatory marker (IL-6) secretion (Appendix A), reduces the metabolic activity of viable cells (Appendix A) and induces ICAM-1 expression (Appendix A). Taken together, these results indicate that GYY4137 enhances intracellular H_2_S content and TNF-α cause endothelial dysfunction in HUVECs.

Endothelial dysfunction causes loss of integrity of the vascular endothelium and affects angiogenesis. To investigate whether angiogenesis is influenced by GYY4137, the HUVECs were cultured on Matrigel under four experimental groups (Figure 2A). Branch formation, as indicated by the presence of capillary-like structures, was found in the HUVECs cultured in medium (control) and GYY4137 group. Limited branch formation was detected in the HUVECs treated with TNF-α (Figure 2B) (*p* < 0.05) and was rescued by the addition of GYY4137. The number of junctions was significantly increased in the presence of GY4137 in comparison to the control group (Figure 2C) (*p* < 0.05).

Next, the role of GYY4137 on apoptosis in TNF-α-stimulated HUVECs was investigated. Annexin V/PI-positive staining showed increased cell apoptosis in the presence of TNF-α compared to the control (*p* < 0.05). GYY4137 post-TNF-α treatment significantly decreased apoptosis in TNF-α-treated HUVECs (*p* < 0.05) (Figure 3A,B) and decreased IL-6 release (Appendix A). To investigate whether these protective effects occur via the intrinsic apoptotic pathway, Cyto c release and downstream caspase activity were analysed. Figure 3C shows that TNF-α significantly increased caspase 3/7 activity compared to untreated cells (*p* < 0.05). Post-treatment with GYY4137 significantly reduced caspase activity in TNF-α-treated cells (*p* < 0.001). As shown in Figure 3D, TNF-α treatment increased the expression of cytosolic Cyto c (*p* < 0.05), whilst GYY4137 post-treatment reduced Cyto c release in TNF-α-treated cells (*p* < 0.05). To further analyse the downstream cascade signalling regulated by TNF-α, protein expression of pro-caspase 3 and cleaved caspase 3 were tested by immunoblotting. However, the levels of caspase 3 were not different between the groups (Figure 3E). On the other hand, cleaved caspase 3 protein expression was increased in the presence of TNF-α, which was significantly decreased with GYY4137 post-treatment (Figure 3F). These results suggested that GYY4137 may attenuate apoptosis via an intrinsic caspase-dependent pathway.

### 3.2. Exogenous H_2_S Enhanced S-Sulfhydration of Caspase 3 in Endothelial Cells

Recent reports suggest protein S-sulfhydration may modulate the apoptotic pathway [24]. To investigate whether the reduction in apoptosis and caspase 3/7 activation by GYY4137 was associated with S-sulfhydration, protein levels of sulfhydrated caspase 3 were investigated (Figure 4A). GYY4137 increases total S-sulfhydration proteins in endothelial cells (Appendix A). S-sulfhydrated pro-caspase 3 protein levels were significantly increased in those cells exposed to GYY4137 post-treatment in the presence (*p* < 0.01) or absence of TNF-α (*p* < 0.05) compared to untreated cells (Figure 4B). TNF-α alone did not alter pro-caspase sulfhydration, but a significant increase in sulfhydrated pro-caspase 3 protein expression upon GYY4137 treatment (*p* = 0.01) was observed. These results suggest that GYY4137 may enhance the S-sulfhydration of proteins, including caspase 3, irrespectively of the pro-inflammatory status.

### 3.3. Exogenous H_2_S Restores the Antioxidant Gene Response and Mitosox Oxidation in TNF-α Treated Endothelial Cells

H_2_S is known to protect vascular endothelial function and prevent apoptosis through its antioxidant properties [9]. To investigate antioxidant gene response, the expression of the antioxidant genes, thioredoxin-1 (Trx1), heoxygenase-1 (HO-1) and mitochondrial superoxide dismutase 2 (SOD2) was analysed. Compared to untreated HUVECs, GYY4137 post-treatment significantly increased antioxidant Trx1 and HO-1 mRNA levels in the presence of GYY4137 post-treatment compared to TNF-α treatment alone (*p* < 0.05; Figure 5A,B). Mitochondrial SOD (SOD2) mRNA expression was significantly elevated upon TNF-α treatment alone and with GYY4137 post-treatment compared to untreated cells (*p* < 0.05; Figure 5C).

Next, we compared mitochondrial ROS levels using the MitoSOX probe. HUVECs exposed to TNF-α had significantly higher MitoSOX oxidation (*p* < 0.001), which was attenuated by GYY4137 post-treatment (*p* < 0.001) (Figure 6A,B). A similar mitochondrial ROS attenuation was detected by fluorescence microscopy (Appendix A) and using the intracellular ROS probe, CM-H2DCFDA (Appendix A). These results suggest that GYY4137 post-treatment can regulate TNF-α-induced oxidative stress by abrogating mitochondrial ROS and enhancing the expression of antioxidant genes in endothelial cells.

### 3.4. Exogenous H_2_S Improves Mitochondrial Δψm in Endothelial Cells

The reduction in mitochondrial superoxide production and Cyto c release by GYY4137 post-treatment led us to postulate that H_2_S could attenuate TNF-α impaired mitochondrial function. We then analysed the impact of GYY4137 on mitochondrial membrane potential (Δψm) as an indicator of mitochondrial function using the fluorescent cationic JC-1 dye (Figure 7A). As a cationic dye, JC-1 accumulates in the energised mitochondria, where healthy cells form dye complexes known as J-aggregates and emit red fluorescence. In contrast, in unhealthy mitochondria, JC-1 remains in a J-monomeric form that exhibits green fluorescence [25]. Our results show that TNF-α treatment significantly decreased (*p* < 0.05) the red (~590 nm) to green (~529 nm) fluorescence intensity ratio, indicating impaired and depolarisation of Δψm in HUVECs (Figure 7B). In addition, GYY4137 post-treatment abrogated mitochondrial depolarisation compared to HUVECs treated with TNF-α alone (*p* < 0.01), suggesting that exogenous H_2_S might regulate mitochondrial function via restoring mitochondrial polarisation.

### 3.5. Exogenous H_2_S Improved Mitochondrial Morphology in TNF-α-Treated Endothelial Cells

Mitochondria are highly dynamic organelles that constantly undergo fusion and fission cycles that regulate mitochondrial morphology. Loss of mitochondrial Δψm is reported to modify mitochondrial morphology leading to fragmentation [26]. Therefore, we investigated whether exogenous GYY4137 attenuates TNF-α-induced mitochondrial morphology changes in HUVECs (Figure 8A). Quantification of the changes in mitochondrial morphology, including length and the mitochondrial network, was conducted using the ImageJ macro; Mitochondrial Network Analysis (MiNA) toolset, as described previously [27]. HUVECs challenged with TNF-α exhibited significantly shorter branch lengths (*p* < 0.001) when compared to untreated cells (Figure 8B). Mitochondrial network size, defined as the number of branches in each mitochondria network, was also reduced in the presence of TNF-α compared to untreated cells (*p* < 0.001) (Figure 8C). GYY4137 post-treatment improved the length of branches and mitochondrial network numbers in TNF-α-treated HUVECs (*p* < 0.001). However, regarding the length of branches, the recovery by GYY4137 was partial as the length in TNF-α-treated HUVECs remained significantly reduced compared to untreated cells (*p* < 0.001).

While fusion and fission dynamics are balanced under basal conditions, oxidative stress can provoke a shift towards fission, resulting in excessive mitochondrial fragmentation [28]. qPCR analysis shows that mitochondrial fission marker DRP1 mRNA levels were significantly increased in TNF-α treated cells compared to untreated (*p* < 0.01), and that GYY4137 post-treatment significantly revoked this rise (*p* < 0.05) (Figure 9A). Furthermore, Figure 9B shows that GYY4137 post-treatment significantly elevated mitochondrial fusion marker, MFN1 mRNA expression (*p* < 0.01). Taken together, these results suggest that TNF-α affects mitochondrial morphology, leading to overall smaller networks and shorter branches, whilst post-treatment with GYY4137 could improve but does not completely restore mitochondrial morphology in TNF-α treated HUVECs.

## 4. Discussion

This study revealed that exogenous H_2_S plays a protective role against endothelial dysfunction upon GYY4137 post-treatment via regulation of the intrinsic apoptotic pathway, abrogating mitochondrial dysfunction, and enhancing S-sulfhydration of caspase 3. H_2_S has been reported to exhibit a wide range of physiological functions, including blood vessel relaxation and cardioprotection [10,29,30]. Most in vitro studies that investigated these protective roles of H_2_S have used pre-treatment or co-treatment approaches [16,17,19,31]. For example, pre-treatment of H_2_S protects cardiomyocytes against apoptosis in H_2_O_2_-induced oxidative stress through the activation of antioxidant enzymes such as SOD [31]. Even though pre- and co-treatment allows investigation of the prevention or occurrence of disease, post-treatment offers the benefit of evaluating the suitability of a drug as a treatment in a disease condition [32]. Previous work in our laboratory showed that the dysfunctional cystathionine γ-lyase (CSE)/H_2_S pathway is a contributor to the pathogenesis of preeclampsia using a C57BL/6 mice model [33]. GYY4137 treatment restored foetal growth and the placental vasculature compromised by the CSE inhibitor, DL-propargylglycine. In addition, GYY4137 improved blood pressure, liver function, and foetal weight in mice where these parameters were compromised by DL-propargylglycine treatment. Liu reported that GYY4137 decreased vascular inflammation and oxidative stress, improved endothelial function and reduced atherosclerotic plaque formation in ApoE−/− mice [34]. These results suggested that endogenous H_2_S is important for healthy vasculature to support well-being. Following in vivo work, we investigated molecular mechanisms underlying protective effects exerted by H_2_S slow-releasing donor, GYY4137.

Even though TNF-α is commonly known to induce extrinsic apoptotic pathways, previous studies reported that TNF-α could play a dual role by mediating extrinsic and intrinsic apoptotic pathways [22,35]. In line with these reports, our results also show that TNF-α increases Cyto c release, leading to mitochondrial-mediated intrinsic apoptosis pathway activation. TNF-α reduced the Δψm and enhanced ROS formation, which has previously been reported to lead to the loss of Δψm in HeLa cells [36]. The imbalance of Δψm has been reported to trigger structural modifications in the organelle from tubular to globular mitochondria [37], which can alter the expression or activity of fission and/or fusion proteins and, ultimately, contribute to mitochondrial functional impairment. Relatedly, Rao and colleagues demonstrated that silencing human cystathionine beta-synthase (CBS) can cause mitochondrial fragmentation in HUVECs, which can be ameliorated by GYY4137 treatment [12]. Our results suggest that H_2_S may contribute to mitochondrial membrane remodelling and maintaining mitochondrial heath in endothelial cells.

The loss of Δψm can subsequently account for Cyto c release followed by caspase activation. Apart from this intrinsic pathway, caspase 3 could also be activated via extrinsic pathway induced by TNF-α treatment [23]. As a pro-apoptotic enzyme, caspase 3 activity has been associated with vascular dysfunction in vivo, and its increased activity can be ameliorated by H_2_S [38]. Previously, the protective effect of H_2_S post-treatment against TNF-α-mediated Cyto c release and caspase 3 protein expression in astrocytes was reported [39]. Our results demonstrate that GYY4137 post-treatment could decrease Cyto c release and reduce caspase 3 activation.

Previous studies reported that Cys-163 was the active catalytic site of caspase-3 [40]. A subset of pro caspase-3 is known to be S-nitrosylated at the active catalytic site cysteine in resting lymphocytes [41,42]. During Fas-induced apoptosis, caspases are denitrosylated at the catalytic site. Glutathiolation also been described as another regulatory mechanism in caspase 3. It was shown that an increase in caspase 3 S-glutathiolation attenuated cleavage, resulting in the inhibition of TNF-α-mediated endothelial cell death [43]. Recent work has suggested pro-caspase-3 can be constitutively S-sulfhydrated at Cys-163 using fast-releasing H_2_S donors [24]. However, these effects have not been previously explored using slow-releasing H_2_S donors in endothelial cells. Here, we demonstrate for the first time that the slow-releasing donor GYY4137 can also S-sulfhydrated caspase 3 in endothelial cells, supporting the assumption that S-sulfhydration may act as a safeguard mechanism in endothelial cell death. Previous analysis in HeLa cells with H_2_S fast-releasing prodrugs suggested that conformational modification during caspase 3 maturation, exposing catalytic thiol, may lead to higher S-sulfhydration in cleaved caspase 3 compared to inactive pro-caspase forms [24].

Furthermore, it can be speculated that the reduction in apoptosis may also be associated with the S-sulfhydration of other players, such as Kelch-like ECH-associated protein 1 (Keap1). S-sulfhydration of Keap1 facilitates nuclear translocation of nuclear factor erythroid 2–related factor 2 (Nrf2), which induces the expression of a battery of antioxidant genes triggering an anti-apoptotic response, including HO-1 [44,45].

There are some inherent limitations in the selective detection of S-sulfhydration by biotin-switch assay due to the reactivity of the persulfide group (RSSH) with other sulfur species such as thiols. The modified biotin switch technique, using MMTS as an alkylating reagent, overcomes many of these unspecific reactions, but it is still possible that not all free thiols are blocked during the MMTS labelling step [46]. More sensitive and specific methods, such as tandem mass spectrometry, which allows direct and unbiased proteomic mapping of s-sulfhydration, would enable confirmation of both the occurrence and location of sulfhydration. Several recent studies reported the proteomic mapping of S-sulfhydration in a broad range of cell types [21,47]. Bibli et al., 2021 performed proteomic analysis to map the S-sulfhydrome of endothelial cells isolated from human arteries [48]. Their study revealed that short-term H_2_S supplementation increased protein s-sulfhydration and improved vascular function. Future studies to perform proteomic analysis in the presence of slow-releasing H_2_S donors could be undertaken to map the GYY4137-mediated changes to sulfhydrome. Identifying specific cysteine residues prone to sulfhydration can then be targeted for site-directed mutagenesis studies to confirm their roles. Another limitation is that the present in vitro model is based on one type of primary endothelial cells. Therefore, this culture microenvironment does not reflect the physiological characteristics of an endothelium. Development of 3D cultures of endothelial cells or co-culture models with smooth muscle cells and pericytes may provide more information about how H_2_S donors will improve vascular health. Furthermore, the use of in vivo models will help to understand overall vascular and body health related to H_2_S donors.

In conclusion, our results demonstrate that exogenous slow-releasing H_2_S donor, GYY4137 inhibits downstream apoptotic intrinsic mechanisms through S-sulfhydration of caspase 3, leading to downregulation of its activity. Furthermore, increased intracellular H_2_S levels were cytoprotective and anti-apoptotic effects were associated with increased antioxidant gene expression, reducing ROS and improving mitochondrial health.

## Figures and Tables

**Figure 1 antioxidants-12-00734-f001:**
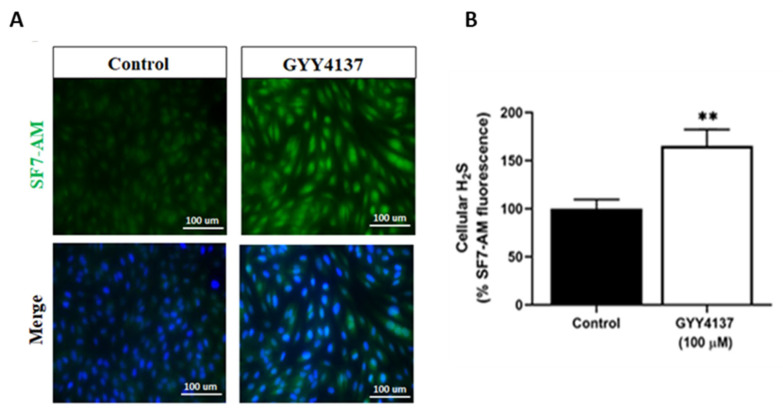
GYY4137 induces H_2_S in HUVECs. (**A**) Intracellular H_2_S content in HUVECs was evaluated by double staining with SF7-AM (an H_2_S marker, green) and DAPI (a nuclei marker, blue). Scale bar 100 µm. (**B**) Quantification of fluorescence intensity in GYY4137-treated cells relative to untreated HUVECs (control). A total of 100 cells per field in each group per independent experiment were analysed using DAPI staining, and SF7-AM fluorescence intensity was measured. *p*-values were calculated using the unpaired Student *t*-test. Data are shown as means ± SD; ** *p* < 0.01 vs. control, (n = 3).

**Figure 2 antioxidants-12-00734-f002:**
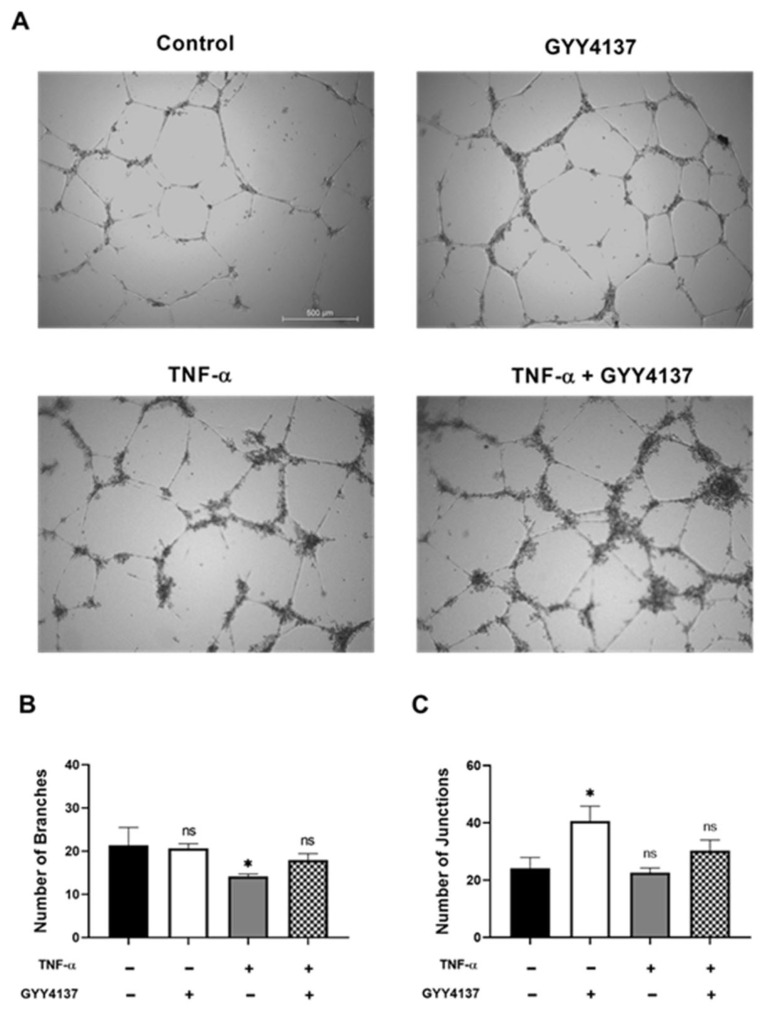
Effects of GYY4137 in endothelial angiogenesis. HUVECs were treated with TNF-α (1 ng/mL, 3 h) followed by GYY4137 (100 μM, 21 h). (**A**) Representative images of vessel formation by HUVECs. A Nikon microscope captured images using 4× objective magnification (scale bar: 500 μm). (**B**) The number of branches and (**C**) the number of junctions was analysed by ImageJ software. Results were shown as mean ± SD, 3 independent experiments with 3 replicate in each experiment. Significance was determined by a one-way ANOVA test followed by Tukey’s post-test comparing treatments to control. ns. (non-significant), * *p* < 0.05 compared to control.

**Figure 3 antioxidants-12-00734-f003:**
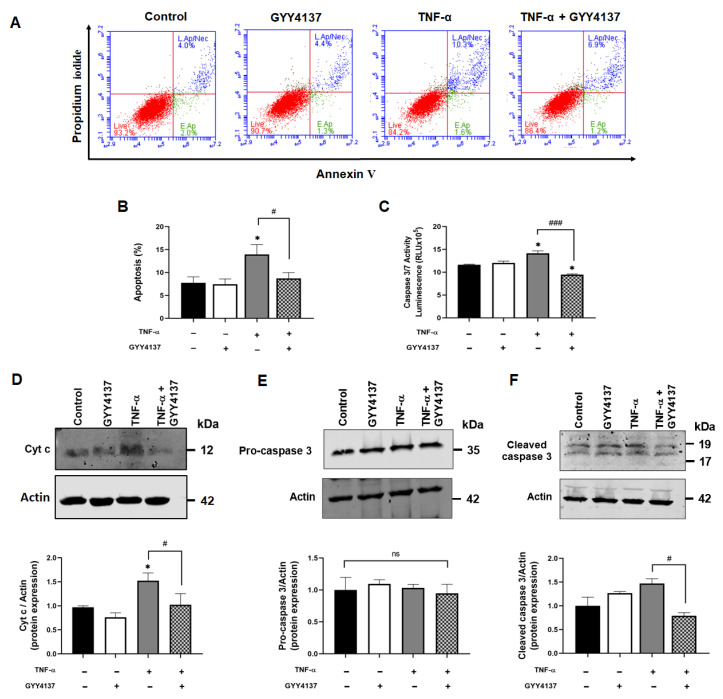
GYY4137 post-treatment arrests intrinsic apoptotic pathway. (**A**) Representative scatter plots showing apoptosis/necrosis (%) detected by Annexin V/Propidium Iodide double-staining with 10,000 events in each replicate. (**B**) Apoptosis (%) in treated HUVECs determined by flow cytometry from three independent experiments (**C**) The activity of Caspase 3/7 was analysed using a Promega^®^ luminescence assay kit from three independent experiments. (**D**) Western blot analysis for cytosolic cytochrome c (Cyto c) (**E**) pro-caspase 3 and (**F**) cleaved caspase 3 protein expression. *p*-values were calculated using one-way ANOVA test followed by Tukey’s comparison test. Data are shown as means ± SD; * *p* < 0.05 vs. control. ^#^
*p* < 0.05 and ^###^
*p* < 0.001 vs. TNF-α-treated groups only. N = 3 independent experiments with 3 replicate in each experiment.

**Figure 4 antioxidants-12-00734-f004:**
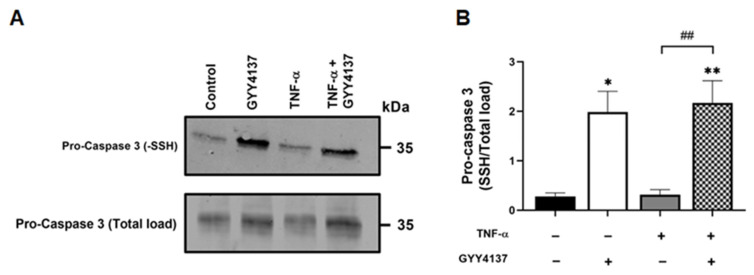
GYY4137 post-treatment mediates sulfhydration of pro-caspase 3. (**A**) Representative immunoblots of S-sulfhydrated and total load pro-caspase 3 protein expression in HUVECs. (**B**) Bar graph of S-sulfhydrated pro-caspase 3 (-SSH/Total load) quantified by ImageJ. *p*-values were calculated using one-way ANOVA test followed by Tukey’s comparison test. Data are shown as means ± SD; * *p* < 0.05, ** *p* < 0.01 vs. control, ^##^
*p* < 0.01 vs. TNF-α-treated groups only. N = 3 independent experiments with 3 replicate in each experiment.

**Figure 5 antioxidants-12-00734-f005:**
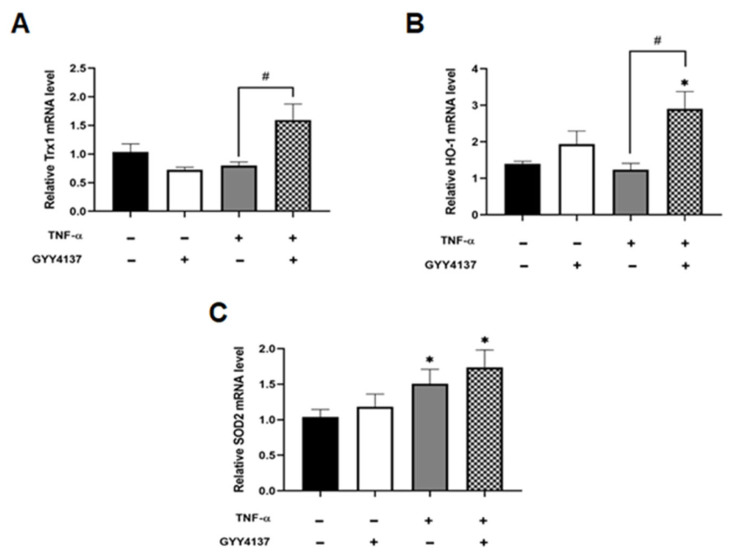
GYY4137 post-treatment increase intracellular antioxidants in TNF-α-treated cells. The extracted RNA was reverse transcribed to cDNA and calculated using the ΔΔCt value method with two duplicates and with EEF2 as a housekeeping gene. Bar graphs of relative (**A**) Trx1, (**B**) HO-1, and (**C**) SOD2, mRNA expression was analysed by RT-QPCR. Results were shown as mean ± SD (N = 4 independent experiments with 3 replicates in each experiment. One way ANOVA test was carried out, followed by Tukey’s post hoc test to normalise for multiple comparisons comparing treatment vs. control, whereby (* *p* < 0.05) compared to control. (# *p* < 0.05) represents a comparison to the TNF-α group alone.

**Figure 6 antioxidants-12-00734-f006:**
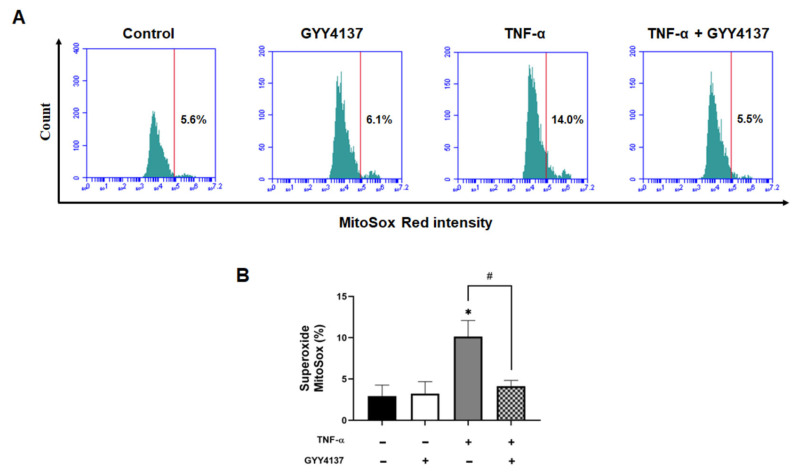
GYY4137 post-treatment reduces oxidative stress in TNF-α-treated cells. (**A**) Representative flow cytometry histogram plots for MitoSox intensity and (**B**) quantification of MitoSox fluorescence-associated superoxide intensity compared to the untreated control (positive threshold was set to 2% using unlabelled cells), using 10,000 events in each replicate *p*-values were calculated using one-way ANOVA followed by Tukey’s comparison test. Data are shown as means ± SD; * *p* < 0.05 vs. control. # *p* < 0.05 vs. TNF-α-treated groups only. N = 3 independent experiments with 3 replicate in each experiment.

**Figure 7 antioxidants-12-00734-f007:**
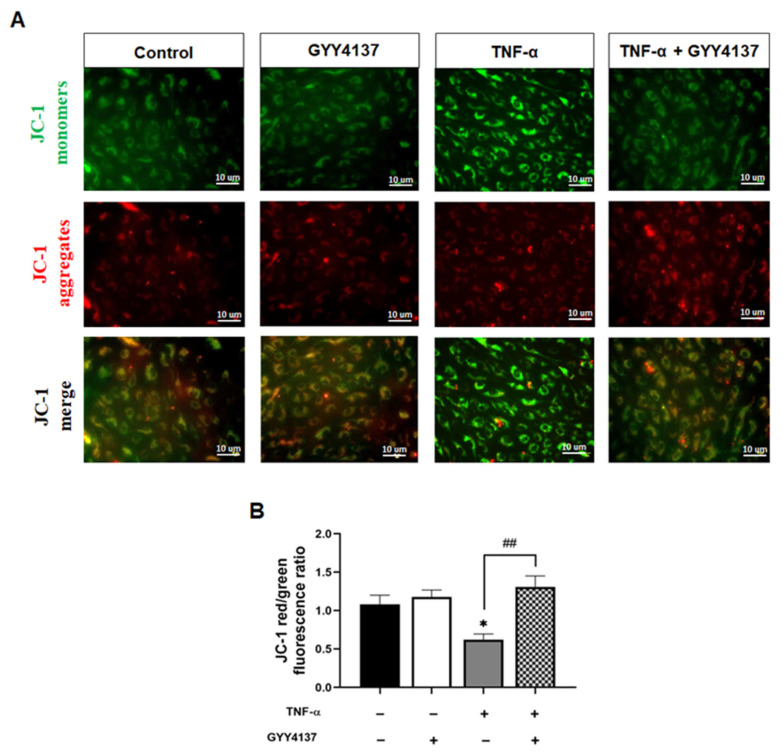
GYY4137 post-treatment prevents mitochondrial depolarisation mediated by TNF-α. (**A**) Representative fluorescence images showing mitochondrial membrane potential (Δψm) detected by JC-1 staining. Scale bar 10 µm. A decrease in the red (~590 nm)/green (~529 nm) fluorescence intensity ratio is indicative of depolarisation (n = 3). (**B**) Quantifying Δψm in the presence of TNF-α with GYY4137post-treatment in 4 fields with at least 40 cells was monitored. *p*-values were calculated using one-way ANOVA test followed by Tukey’s comparison test. Data are shown as means ± SD * *p* < 0.05 vs. control. ^##^
*p* < 0.01 vs. TNF-α-treated groups only. N = 3 independent experiments with 3 replicate in each experiment.

**Figure 8 antioxidants-12-00734-f008:**
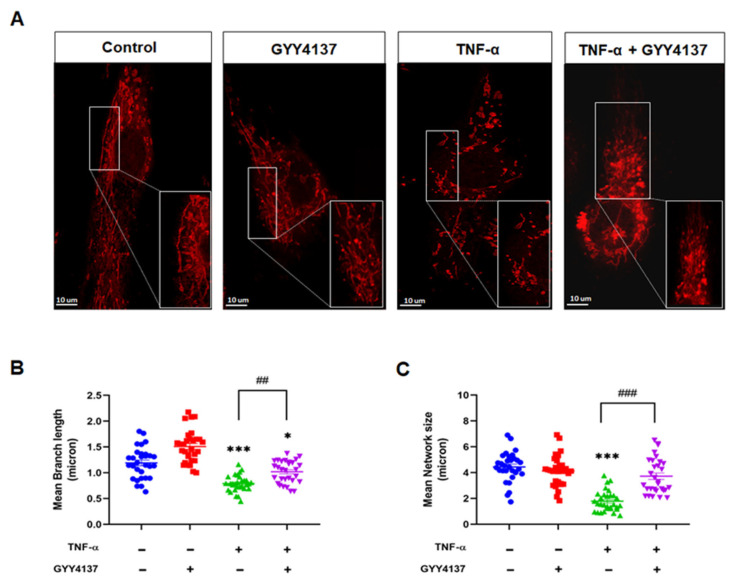
GYY4137 post-treatment prevents mitochondrial morphology changes mediated by TNF-α. (**A**) Mitochondria morphology was determined by Mitotracker (red staining, 100 nM) and confocal microscopy imaging (63× oil APO objective lens). Images of isolated mitochondrial networks were analysed using a publicly available ImageJ macro, MINA [24]. Scale bar 10 µm. (**B**) Mitochondria mean branch length is the average length (in microns) of all branches. (**C**) Mitochondria network size is defined as the average number of branches per network (in microns). *p*-values were calculated using one-way ANOVA test followed by Tukey’s comparison test. A total of 30 cells per treatment group per independent experiment were consolidated and measured. Data are shown as means ± SD; * *p* < 0.05, *** *p* < 0.001 vs. control. ^##^
*p* < 0.01, ^###^
*p* < 0.001 vs. TNF-α-treated groups only. N = 4 independent experiments with 3 replicate in each experiment.

**Figure 9 antioxidants-12-00734-f009:**
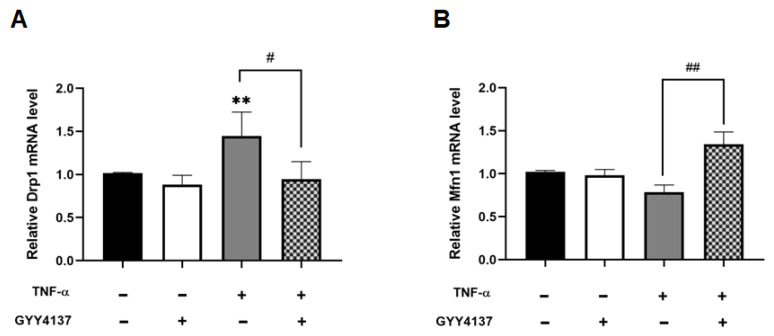
GYY4137 post-treatment ameliorates disruption of the mitochondrial fission/fusion machinery mediated by TNF-α. (**A**) Analysis of pro-fission gene showing mRNA levels of DRPI1 N = 6 independent experiments with 3 replicates in each experiment. (**B**) RT-qPCR test of pro-fusion mRNA expression of MFN1. The extracted RNA was reverse transcribed to cDNA and calculated using the ΔΔCt value method with EEF2 as the housekeeping gene (n = 9). Results were shown as mean ± SD. One way ANOVA test was carried out followed by Sidak’s post hoc test to normalise for multiple comparisons comparing treatment vs. control, whereby (**) represents *p* < 0.01 compared to control. (# *p* < 0.05) and (## *p* < 0.01) represents a comparison to the TNF-α group alone.

## Data Availability

The data presented in this study are available in the article and Appendix A.

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
