# Peer review of "TNF-α-Mediated Endothelial Cell Apoptosis Is Rescued by Hydrogen Sulfide"

_antioxidants, 2023, doi:10.3390/antiox12030734_

Round 1

Reviewer 1 Report (Previous Reviewer 1)

The most comments and questions have been satisfactorily answered by the authors.

The following points should still be adjusted.

Reviewer 1:Research articles using only one type of cell line for the experiments are not convincing, since it is not clear whether these are general reported effects or just an artefact of the used single cell line/model. Therefore, the reported key experiments need to be repeated on 1-2 more cell lines or primary isolated cells to clearly demonstrate the general importance of the provided the results.

 Answer authors: Endothelial cells build the inner layer of blood and lymphatic vessel and control the passages of substances and cells between the blood and the neighbouring tissues. Therefore, in this paper we were interested in understanding molecular mechanisms by which GYY4137 protect endothelial cells. We have first used an endothelial cell line, EA.hy926 to test our hypothesis (Figure 1). We observed GYY4137 (100μM) increases intracellular H2S levels and IL-6 secretion. However, we did not observe changes to CM-H2DCFDA probe detected H2O2 levels. Primary endothelial cell cultures are superior in their physiological relevance compared to immortalised cell lines. Hence, we continued to investigate primary endothelial cell line, HUVECs to identify benificial effects of H2S on vasculature.

 Answer reviewer1:

The authors should include a limitation statement at the end of the discussion section. Here, it should be clearly stated that the provided results were generated from one type of primary endothelial cell line.

7. Reviewer 1:Methods: From the description of the experiment set up it is not clear if or how controls were treated. Did authors use placebo, vehicle, solvent etc?

 Answer authors : Controls were treated with cell culture media. This is now described in methods, page 6

 Answer reviewer1:

For this reviewer it is not clear if the authors used the correct treatment of the controls. It is not described in the methods in what kind of solvent the GYY4137 is solved. GYY4137 is often solved in DMSO. Then the controls should be treated with the same amount of DMSO. Please clearly describe the performed experiment.

 9. Reviewer 1:Figure legends: In addition, please clearly describe in the corresponding figure legends the number of performed independent experiments and how many replicates each independent experiment contains.

 Answer authors : We acknowledge this comment. We have updated all figure legend to indicate how many replicates each independent experiment contains.

 Answer reviewer1:

The figure legends have not been updated. Please update as mentioned.

For example

original legend:

Figure 2. GYY4137 post-treatment arrests intrinsic apoptotic pathway. (A) Representative scatter plots showing early and late apoptosis (%) detected by Annexin V/Propidium Iodide double-stain ing. (B) Apoptosis (%) in treated HUVECs determined by flow cytometry. (C) The activity of  Caspase 3/7 was analysed using Promega® luminescence assay kit (D) Western blot analysis for cytosolic cytochrome c (Cyt c) (E) pro-caspase 3 and (F) cleaved caspase 3 protein expression. P-values were calculated were calculated using one-w_a_y_ _A_N_O_V_A_ _t_e_s_t_ _f_o_l_l_o_w_e_d_ _b_y_ _T_u_k_e_y_’s_ _c_o_m_p_a_r_i_-_son test. Data are shown as means ± SD; *p<0.05 vs. control. #p<0.05 and ###p<0.001 vs. TNF-α-treated 271 groups only. (n=3).

Revised version:

Figure 3. GYY4137 post-treatment arrests intrinsic apoptotic pathway. (A) Representative scatter plots showing apoptosis/necrosis (%) detected by Annexin V/Propidium Iodide double-staining. (B) Apoptosis (%) in treated HUVECs determined by flow cytometry. (C) The activity of Caspase 3/7 was analysed using Promega® luminescence assay kit (D) Western blot analysis for cytosolic cyto- chrome c (Cyto c) (E) pro-caspase 3 and (F) cleaved caspase 3 protein expression. P-values were calculated were calculated using one-w_a_y_ _A_N_O_V_A_ _t_e_s_t_ _f_o_l_l_o_w_e_d_ _b_y_ _T_u_k_e_y_’s_ _c_o_m_p_a_r_i_s_o_n_ _t_e_s_t_._ _D_a_t_a_ _are shown as means ± SD; *p<0.05 vs. control. #p<0.05 and ###p<0.001 vs. TNF-α-treated groups only. (n=3).

In addition:

-       Methods: Please provide company names for all used products (for example line 82 EGM2 media, line 85 TNF-a, line 86 GYY4137)

Author Response

Reviewer 1

The following points should still be adjusted.

Reviewer 1:Research articles using only one type of cell line for the experiments are not convincing, since it is not clear whether these are general reported effects or just an artefact of the used single cell line/model. Therefore, the reported key experiments need to be repeated on 1-2 more cell lines or primary isolated cells to clearly demonstrate the general importance of the provided the results.

 Answer authors: Endothelial cells build the inner layer of blood and lymphatic vessel and control the passage of substances and cells between the blood and the neighbouring tissues. Therefore, in this paper we were interested in understanding molecular mechanisms by which GYY4137 protect endothelial cells. We have first used an endothelial cell line, EA.hy926 to test our hypothesis (Figure 1). We observed GYY4137 (100μM) increases intracellular H2S levels and IL-6 secretion. However, we did not observe changes to CM-H2DCFDA probe detected H2O2 levels. Primary endothelial cell cultures are superior in their physiological relevance compared to immortalised cell lines. Hence, we continued to investigate primary endothelial cell line, HUVECs to identify beneficial effects of H2S on vasculature.

 Answer reviewer1:

The authors should include a limitation statement at the end of the discussion section. Here, it should be clearly stated that the provided results were generated from one type of primary endothelial cell line.

RESPONSE: Thank you for your comment; we added this limitation to our discussion on lines 500 to 504 as follows:

The present in vitro model is based on one type of primary endothelial cells. Therefore, this culture microenvironment does not fully reflect physiological characteristics of the endothelium. Development of 3D cultures of ECs or co-culture models with smooth muscle cells and pericytes may provide more information about how H2S donors may improve vascular health.

  1. Reviewer 1:Methods: From the description of the experiment set up it is not clear if or how controls were treated. Did authors use placebo, vehicle, solvent etc?

 Answer authors : Controls were treated with cell culture media. This is now described in methods, page 6

 Answer reviewer1:

For this reviewer it is not clear if the authors used the correct treatment of the controls. It is not described in the methods in what kind of solvent the GYY4137 is solved. GYY4137 is often solved in DMSO. Then the controls should be treated with the same amount of DMSO. Please clearly describe the performed experiment.

RESPONSE: This is an important point. GYY4137 ((SML2470, Sigma-Aldrich) is a water-soluble compound. Therefore, we dissolved GYY4137 in EGM media and EGM only media as our controls. We added this detail in methods, lines 87-89

  1. Reviewer 1:Figure legends: In addition, please clearly describe in the corresponding figure legends the number of performed independent experiments and how many replicates each independent experiment contains.

 Answer authors : We acknowledge this comment. We have updated all figure legend to indicate how many replicates each independent experiment contains.

 Answer reviewer1:

The figure legends have not been updated. Please update as mentioned.

For example

original legend:

Figure 2. GYY4137 post-treatment arrests intrinsic apoptotic pathway. (A) Representative scatter plots showing early and late apoptosis (%) detected by Annexin V/Propidium Iodide double-stain ing. (B) Apoptosis (%) in treated HUVECs determined by flow cytometry. (C) The activity of  Caspase 3/7 was analysed using Promega® luminescence assay kit (D) Western blot analysis for cytosolic cytochrome c (Cyt c) (E) pro-caspase 3 and (F) cleaved caspase 3 protein expression. P-values were calculated were calculated using one-w_a_y_ _A_N_O_V_A_ _t_e_s_t_ _f_o_l_l_o_w_e_d_ _b_y_ _T_u_k_e_y_’s_ _c_o_m_p_a_r_i_-_son test. Data are shown as means ± SD; *p<0.05 vs. control. #p<0.05 and ###p<0.001 vs. TNF-α-treated 271 groups only. (n=3).

Revised version:

Figure 3. GYY4137 post-treatment arrests intrinsic apoptotic pathway. (A) Representative scatter plots showing apoptosis/necrosis (%) detected by Annexin V/Propidium Iodide double-staining. (B) Apoptosis (%) in treated HUVECs determined by flow cytometry. (C) The activity of Caspase 3/7 was analysed using Promega® luminescence assay kit (D) Western blot analysis for cytosolic cyto- chrome c (Cyto c) (E) pro-caspase 3 and (F) cleaved caspase 3 protein expression. P-values were calculated were calculated using one-w_a_y_ _A_N_O_V_A_ _t_e_s_t_ _f_o_l_l_o_w_e_d_ _b_y_ _T_u_k_e_y_’s_ _c_o_m_p_a_r_i_s_o_n_ _t_e_s_t_._ _D_a_t_a_ _are shown as means ± SD; *p<0.05 vs. control. #p<0.05 and ###p<0.001 vs. TNF-α-treated groups only. (n=3).

RESPONSE: Thank you- Additional information has now been added. Line 291

In addition:

-       Methods: Please provide company names for all used products (for example line 82 EGM2 media, line 85 TNF-a, line 86 GYY4137)

RESPONSE: We have now included company names and catalogue numbers of the compounds used.

Reviewer 2 Report (New Reviewer)

It was shown that the protective effect of NaHS against HHcy induced mitochondrial toxicity and endothelial dysfunction in bEND3 cells (19). In this manuscript, the authors investigated the molecular mechanism H2S protects endothelial cells from TNF-α-mediated mitochondrial dysfunction and apoptosis during inflammation.

Using a slow H2S releasing donor, GYY4137 and human umbilical vein endothelial cells (HUVECs), they showed that

1)   Intracellular H2S content in HUVECs was evaluated by staining of SF7-AM (figure 1),

2)   The limited angiogenesis determined by the branch formation in the HUVECs treated with TNF-α was rescued by the addition of GYY4137 (figure 2),

3)   GYY4137 post-treatment arrested intrinsic apoptotic pathways analyzed by annexin V/PI-positive staining, Caspase 3/7 activity, cytosolic cytochrome c and pro-caspase 3 and cleaved caspase 3 protein expression (figure 3).

4)   GYY4137 enhanced S-sulfhydration of caspase 3 in the HUVECs treated with or without TNF-α (figure 4).

5)   GYY4137 increased Trx1, HO-1, and SOD2 mRNA expression in the HUVECs treated with or without TNF-α (figure 5).

6)   Increasing mitochondrial ROS (MitoSox intensity) in the HUVECs treated with TNF-α was attenuated by the addition of GYY4137 (figure 6).

7)   Decreasing mitochondrial membrane potential (JC-1 intensity) in the HUVECs treated with TNF-α was attenuated by the addition of GYY4137 (figure 7).

8)   Shorter branch length and reduced number of branches in each mitochondrial network in the HUVECs treated with TNF-α was restored by the post-addition of GYY4137 (figure 8).

9)   Increasing mitochondrial fission marker, DRP1 mRNA levels in the HUVECs treated with TNF-α was attenuated by the addition of GYY4137 (figure 9).

Based on the above results they propose that post-treatment with GYY4137 ameliorated TNF-α-mediated endothelial dysfunction via mitochondrial dysfunction and apoptosis. Although whether S-sulfhydration of caspase 3 at Cys163 is not obscure, I think this work is well organized and a nice contribution to the field of the mechanism by which H2S protects endothelial cells from mitochondrial dysfunction and apoptosis during inflammation.

This work could serve as a basis for further consideration of H2S-signaling molecules as a potential therapeutic target for endothelial dysfunction, associated with a pro-inflammatory phenotype, mitochondrial dysfunction, and imbalance in the cellular redox steady-state.

I have only few concerns to address as below.

1, The sentences between “GYY4137 post-treatment restored the length of branches and mitochondrial network numbers in TNF-α-treated HUVECs (p<0.001)” and “However, the length of branches remained significantly reduced in comparison to untreated cells (p<0.001)” is not compatible (lanes 376-379).

2, The data of Figure S6C does not represent that of figure S6A. In my opinion, the CM-H2DCFDA fluorescence in the cells treated with GYY4137 is most evident in figure S6A but the quantification column is highest in the cells treated with TNF-α in figure S6C.

3, The authors would better off preparing the graphical abstract for readers’ better understandings.

Author Response

Reviewer 2

1, The sentences between “GYY4137 post-treatment restored the length of branches and mitochondrial network numbers in TNF-α-treated HUVECs (p<0.001)” and “However, the length of branches remained significantly reduced in comparison to untreated cells (p<0.001)” is not compatible (lanes 376-379).

RESPONSE: Thank you for your suggestion, this is now corrected, line 386.

2, The data of Figure S6C does not represent that of figure S6A. In my opinion, the CM-H2DCFDA fluorescence in the cells treated with GYY4137 is most evident in figure S6A but the quantification column is highest in the cells treated with TNF-α in figure S6C.

RESPONSE: Thank you for noting this mistake. We can confirm increased of CM-H2DCFDA fluorescence in the presence of TNF-α treatment. We have now corrected figure page S6A.

3, The authors would better off preparing the graphical abstract for readers’ better understandings.

Graphical abstract was created and submitted alongside the manuscript. But just in case it has been attached.

RESPONSE: Thank you. We will also submit graphical abstract separately during the submission

Reviewer 3 Report (New Reviewer)

There are no real issues that need to be addressed. Interesting job. Just some comments may be enclosed on a possible further evaluation on in vivo models (animals) to test if their results may be translated in more complex biological systems

Author Response

Reviewer 3

There are no real issues that need to be addressed. Interesting job. Just some comments may be enclosed on a possible further evaluation on in vivo models (animals) to test if their results may be translated in more complex biological systems

RESPONSE: Thank you for your suggestion- This information has now been added from lines 503-504.

This manuscript is a resubmission of an earlier submission. The following is a list of the peer review reports and author responses from that submission.

Round 1

Reviewer 1 Report

In the manuscript: „TNF-a-mediated endothelial cell apoptosis is rescued by hydrogen sulfide“, the authors aimed to investigate the effects of a posttreatment with a slow-releasing hydrogen sulfide (H2S) donor in a TNF-a-mediated endothelial dysfunction model using a HUVEC cell culture system.

This reviewer has major concerns about the novelty and the study outline as listed in the following sections:

Major points:

-       Research articles using only one type of cell line for the experiments are not convincing, since it is not clear whether these are general reported effects or just an artefact of the used single cell line/model. Therefore, the reported key experiments need to be repeated on 1-2 more cell lines or primary isolated cells to clearly demonstrate the general importance of the provided the results.

-       The novelty of the manuscript is limited to the aspect that the authors used a posttreatment with a slow-releasing H2S donor. There are already publications on this topic. Please see the following publications:

Li-Long Pan et al. Hydrogen Sulfide attenuated tumor necrosis factor-a-induced inflammatory signaling and dysfunction in vascular endothelial cells. PLOS One 2011; 6(5): e19766.

The authors reported that they are presenting the first data of experiments using a slow-releasing donor GYY4137 in contrast to the already published experiments with fast-releasing H2S donors. Here, it would improve the study to expand the experiments with groups using a fast-releasing H2S donor and compare the results directly to the slow-releasing H2S donor to demonstrate the potential differences.

-       To improve the significance of the manuscript, the authors need to include functional endothelial assays such as the aortic ring assay, angiogenesis sprouting assay, matrigel plugs, sponge implants or corneal assays.

-       In order to substantiate the relevance of the results, it would make sense to extend the study for in vivo experiments in mice. It would significantly improve the study whether an atherosclerosis model was used and the effects of GYY4137 were examined in this model. A cell culture model can only hardly display the crosstalk between different cell types in vivo.

-       Methods: The passage number of the used HUVECs is missing, which is an important aspect since the passage number should not exceed more than ten for optimal results (Liao, H., He, H., Chen, Y., Zeng, F., Huang, J., Wu, L., & Chen, Y. (2014). Effects of long-term serial cell passaging on cell spreading, migration, and cell-surface ultrastructures of cultured vascular endothelial cells. Cytotechnology, 66(2), 229–238. https://doi.org/10.1007/s10616-013-9560-8).

-       Important literature using GYY4137 in vivo should be included:

Z. Liu, Y. Han, L. Li, H. Lu, G. Meng, X. Li, et al.

The hydrogen sulfide donor, GYY4137, exhibits anti-atherosclerotic activity in high fat fed apolipoprotein E(-/-) mice;Br J Pharmacol, 169 (2013), pp. 1795-1809

-       Methods: From the description of the experiment set up it is not clear if or how controls were treated. Did authors use placebo, vehicle, solvent etc?

-       Results Figure 1: Here, it is not clear whether the measurement of the cellular H2S has been normalized to DAPI positive cells.

-       Figure legends: In addition, please clearly describe in the corresponding figure legends the number of performed independent experiments and how many replicates each independent experiment contains.

-       Results intrinsic apoptosis pathway: The authors demonstrated that the intrinsic apoptotic pathway is activated. In addition, it is known that the extrinsic pathway is triggered by a death ligand binding to a death receptor like TNF-a to TNFR1. Therefore, the authors need to expand their study for some experiments that analyse the extrinsic pathway like caspase 8 cleavage.

-       Figure 2D: The shown actin blot is problematic. In all other actin blots like 2E or F the authors showed one clear actin band as expected. In 2D, the shown actin band shows a shift in size in the two groups of TNFa and TNF-a+GYY4137 compared to the first two lanes.

-       Figure 3A: The shown immunoblot of pro-caspase 3 total is overexpressed and could not be used for quantification.

-       Figure 4C: The representative western blot image of GcLC did not reflect the presented quantification in 4D.

-       Discussion: The results need to be validated in in vivo experiments to conclude that the slow-releasing H2S donor treatment has therapeutic potential.

Reviewer 2 Report

Diaz Sanchez et. described the effect of hydrogen sulfide on TNF-alpha-induced cell death in HUVEC cells. This is an interesting manuscript, but I have some major concerns:

1)    As HUVEC cells can get senescent during passaging, the authors have to describe which passages of the cells were used for the experiments.

2)    I have problems with the analysis of cell death – apoptosis. From Fig. 2, they cannot distinguish between late apoptosis, as mentioned, and necrosis. Necrosis is defined by damage to the cell membrane and PI-positive staining, and this was shown in Fig. 2A. Within Fig. 2D, there seems to be a problem with the actin staining. It is clearly a double band, which should not exist.

3)    In Fig. 4, I do not see any change in the Western blot of GcLC. For the statement,  restoration of the redox balance, the authors have to analyze SOD by western blotting and not by PCR.

4)    The authors stated that they have analyzed the MitoSox intensities in Fig. 5. This was not the case, as the authors only analyzed the percentage of cells above a threshold.

Reviewer 3 Report

This is an article about the use of GYY as a sulfur donor to repair endothelial cell dysfunction.  For the most part, results and conclusions are overstated for experiments based on in vitro cell culture.

Even though pre- and co-treatment allows investigation of the prevention or occurrence of disease, post-treatment offers the benefit of evaluating the suitability of a drug as a treatment in a disease condition [28]. Not from a cell culture model. Huge leap of logic

“Therefore, the post-treatment experimental approach for H2S presented here provides novel insights into the suitability of GYY4137 as a treatment strategy against endothelial dysfunction”.  GYY has been around as a research tool for years.  I don’t think this is a novel insight.  Is it used for treating anything now? Why hasn’t it been used before?  Later the authors say “reinforces the potential of H2S as a suitable therapeutic approach for vascular conditions”  So not GYY? 

Figure 4 GYY did not increase intracellular antioxidants, it increased the mRNA of intracellular antioxidants.  It did not activate enzymes either.